# Yellowing, Weathering and Degradation of Marine Pellets and Their Influence on the Adsorption of Chemical Pollutants

**DOI:** 10.3390/polym14071305

**Published:** 2022-03-24

**Authors:** Bárbara Abaroa-Pérez, Sara Ortiz-Montosa, José Joaquín Hernández-Brito, Daura Vega-Moreno

**Affiliations:** 1Marine Litter Observatory of Reserva de la Biosfera de Fuerteventura, 35620 Las Palmas, CI, Spain; b.abaroa@observatoriodebasuramarina.com; 2Chemistry Department, Universidad de Las Palmas de Gran Canaria (ULPGC), 35017 Las Palmas, CI, Spain; sara.ortiz101@alu.ulpgc.es; 3Plataforma Oceánica de Canarias (PLOCAN), 35214 Telde, CI, Spain; joaquin.brito@plocan.eu

**Keywords:** microplastic pellets, weathering, degradation, Yellowness Index, Fourier transform infrared spectroscopy, persistent organic pollutants

## Abstract

Marine microplastics (MPs) are exposed to environmental factors, which produce aging, weathering, surface cracking, yellowing, fragmentation and degradation, thereby changing the structure and behavior of the plastic. This degradation also has an influence on the adsorption of persistent organic pollutants over the microplastic surface, leading to increased concentration with aging. The degradation state affects the microplastic color over time; this is called yellowing, which can be quantified using the Yellowness Index (YI). Weathering and surface cracking is also related with the microplastic yellowing, which can be identified by Fourier transform infrared spectroscopy (FTIR). In this study, the degradation state of marine microplastic polyethylene pellets with different aging stages is evaluated and quantified with YI determination and the analysis of FTIR spectrums. A color palette, which relates to the microplastic color and YI, was developed to obtain a visual percentage of this index. The relation with the adsorption rate of persistent organic pollutant over the microplastic surface was also determined.

## 1. Introduction

Marine microplastics (MPs) are plastic fragments which are smaller than 5 mm [1,2] and can be present in the marine environment for long periods of time. Most of them have been in the ocean for between 5 and 25 years [3] and could have been transported thousands of kilometers in respect to the areas where they were originally dumped [4,5,6]. There are essentially two types: primary MPs, which were already manufactured with this size range, and secondary MPs, which have been formed from the fragmentation of larger plastics [7].

Plastics are derived from petrochemical products. Their high durability and plasticity, low cost and ease of manufacture make them widely used in everyday life. An estimated 4.9 billion tons of plastic waste has been dumped into the environment so far [8], making this product the most abundant component of marine litter [9].

Although their degradation is slow, they undergo aging, weathering and fragmentation processes [2]; factors such as ultraviolet irradiation, exposure to oxygen, salinity, physical and chemical properties of the soil, grains of sand and the presence of microorganisms influence the degree of aging of MP [10], with the weathering being more intense under marine conditions [10]. These processes progressively reduce their size while increasing their relative abundance, due to the division and fragmentation of the macroplastic and mesoplastic into smaller-sized MPs [11]. The longer they remain in the ocean, the more fragmentation the plastic undergoes [12], and the greater the change in its chemical composition [10,13,14,15].

The physical–chemical processes that plastic waste undergoes in the marine environment leads to a progressive degradation of the chemical structure of the polymer [13,15,16], causing a modification in the original characteristics and properties of the plastic [12,14,17] and a contamination of the seawater surrounding the marine debris [18].

### 1.1. Microplastic Photo-Oxidation Processes

The weathering of plastic caused by photo-oxidative processes causes changes in its chemical composition and in its physical–chemical properties, such as absorbance and reflection of light [15], which causes degradation on the surface [13,14] and changes in tonality [19].

The degradation and photo-oxidation suffered by MPs depends, among other factors, on the type and composition of the original plastic [13,20], the crystallinity, polymer type and particle size [21], as well as environmental factors, for which it has a high variability, such as the properties of the medium (salinity, pH and ionic strength), the formation of biofilms or the presence of other competing compounds (that is, organic matter) and marine conditions, where the greatest degradation suffers due to the number of influencing parameters [22].

The degradation-state identification is based on their physical characteristics, determined by microscopy [23], and on their chemical characterization, determined through methodologies such as Fourier transform infrared spectroscopy (FTIR) [13,20,24], Raman spectroscopy [25,26,27] or thermal analysis as pyrolysis gas chromatography–mass spectrometry (Py-GC–MS) [23,28,29]. With these methodologies, the increment of hydroxyl and carbonyl groups on different MP pellets can be determined [12,15].

The photo-oxidation of MP modifies its original color and turns it yellow; this is known as the yellowing process [15]. Although the characterization of the color of the MP is usually included in the studies carried out and is an indicator of its aging [19], it is usually based solely on the objectivity of the researcher as it is often carried out by visual identification [30].

This limits the knowledge about the states of degradation, fragmentation, or the estimation of the time in which the MP has been at the marine environment prior to its sampling [25], which is known to possibly be several decades [3]. To circumvent this limitation, various authors suggest using the Yellowness Index (YI) as an objective and quantifiable measurement parameter of the state of yellowing and degradation of microplastic [19].

### 1.2. Yellowness Index (YI)

The YI value (%) is based on the yellowing of the sample and increases according to MP degradation [19]. This value can be measured objectively using a colorimeter [1]. However, not all routine laboratories that analyze marine MP have this type of equipment, and there is also no single standardized protocol for its determination, causing many authors to make a visual evaluation of the color without quantifying the state of degradation of the sample.

Visually, the yellowing suffered by plastic is associated with burns, soiling and general degradation of the product [31]. According to the ASTM D 1925–70 method or the standardized method E313-15e1, the YI is a mathematical expression that allows the colorimetry in solids to be quantified ([32], pp. 1–6).

### 1.3. Sorption of Chemical Pollutants in Marine MP

Persistent chemical pollutants (POPs) are present in many regions of the ocean due to their high persistence and low rates of degradation [33]. This is due to the presence of aromatic rings or aliphatic halogen compounds, which also give them low solubility in water and affinity to lipophilic compounds [34].

Various studies demonstrate the ability of MP to adsorb and accumulate POPs on its surface [22,35,36,37]. MP acts as a transporter and accumulator of these pollutants over their surface through the ocean [38,39].

This capacity varies depending on the state of fragmentation and degradation of MP [40,41], and varies according to the plastic composition [18,22].

Age, the degree of erosion of the plastic and the chemical properties of the pollutant [21] influence the sorption of pollutants; a greater accumulation of POPs in MPs that presents greater aging and YI [30,36,40] has been found, possibly due to increased surface porosity and roughness [36], which increases the surface/volume ratio of MP due to surface cracking [22,42].

Previous work has shown that the aging process increased the surface negative charges of MPs; this increased the electrostatic attraction or repulsion, which improved the adsorption property [17,22].

In addition, although the hydrophobic interactions are abundant [17] with the degradation, the oxidation of the surface increases, which favors the adsorption of hydrophilic analytes [36,42,43]. However, there remain few studies that address the relationship between the degradation of plastics and their rate of adsorption of pollutants.

The arrival of a relevant quantity of microplastic pellets on the coast of the Canary Islands is common [44,45,46]. In this study, various marine MP pellet samples were collected in the region of the Canary Islands (Spain) during 2019. Their specific composition was evaluated by FTIR, and each YI value was quantitatively determined. In addition, an attempt was made to relate the adsorption rates of the POPs on these samples according to their state of aging, weathering and degradation, calculated according to their YI value and FTIR spectrum, to create a value scale that allows the quick identification of YI in the laboratory and determination of its possible impact.

## 2. Materials and Methods

### 2.1. Marine MP Samples

For this study, only primary MP, marine pellets and virgin high-density polyethylene (HDPE) microplastic pellets (CheMondis^®^, Köln, Germany) were used.

Pellet samples were collected on sandy beaches in the region of the Canary Islands (Gran Canaria and Fuerteventura islands, Spain) in 2019. The pellets were selected and collected directly from the sand with gloves and tweezers. Given the objectives of this study, the amount of marine MP present on the beach at the time of sampling was not counted. Other kinds of plastic and secondary MP, such as fragments or mesoplastic, were discarded. To preserve the possible presence of POPs, pellet samples were wrapped in aluminum foil and stored at a temperature of −80 °C until their subsequent analysis [30].

### 2.2. Equipment

To confirm the composition of each pellet, a FTIR spectrometer equipped with an Attenuated Total Reflection module (ATR-FTIR), model Cary 630 (Agilent Technologies^®^, Santa Clara, CA, USA), was used.

The determination of the color and the identification of the YI value of each pellet was carried out using a colorimeter, video spectrocomparator VSC5000 (Foster Freeman^®^, Evesham, UK).

To evaluate the adsorption rate of pollutants in pellets based on their degradation state, POPs were analyzed using GC-MS equipment (Agilent Technologies^®^, Santa Clara, CA, USA), model 7820A and 5977B MSD, with an HP-5MS Ultra Inert 19091S-433UI column, following a methodology for multiresidual analysis [47]. An orbital shaker was used for pellet conditioning (Supelco^®^, Darmstadt, Germany).

### 2.3. Yellowness INDEX (YI) Determination

The YI of marine MP was quantified using coordinates in a CIE three-dimensional color-space diagram (Figure 1) [1]. This system allows the brightness or luminosity of a color to be quantified (determined by Y, also called L); the color varies from black to white, with values from 0 to 100. In addition, the colorimeter records the chromatic components, represented in the CIE through the variables a and b, varying from the range of green (-a) to red (+a) and from blue (-b) to yellow (+b) [1,48]. These chromatic components can also be represented by the *x* and *y* values, determined by the cut of the Cartesian coordinates in the representation of the CIE diagram (Figure 1).

The *YI*, measured in percentage (%), was determined for the marine microplastic following the ASTM E131 method [31] and ([32], pp. 4–9):(1)$$YI\;\left(\%\right)=100\frac{\;C_{X}X-C_{Z}Z}{Y}$$
where *C_X_* and *C_Z_* are constants of values 1.2769 and 1.0592, respectively.

The *X* and *Z* values are calculated using the following equations:$$X=\frac{Y}{y}\;x\;Z=\frac{Y}{y}\;\left(1-x-y\right)$$

The values (*x*, *y*, *Y*) are given by the colorimeter through the CIE diagram (Figure 1). The parameters *X*, *Z* and *YI* are determined through the indicated equations.

### 2.4. Microplastic Degradation Identified by ATR-FTIR Analysis

The factors that most affect the degradation rate of the pellets are, in addition to the exposure time, environmental conditions such as UV light, salinity and oxygen exposure [13]. This photo-oxidative degradation generates variations in the chemical structure of the MP, which increases as the time the marine MPs spend in the ocean increase [15,16].

Polyethylene (PE), together with polypropylene (PP), are the most widely used hydrocarbon polymers [12]. It is a raw material, which is cheap and easy to process in a multitude of products. It has a linear chemical formula: H(CH_2_CH_2_)_n_H; however, when affected by physical–chemical processes such as photo-oxidation in seawater, oxygen atoms are added to the hydrocarbon chain [12].

The PE backbones are built exclusively from single C-C bonds that do not readily undergo hydrolysis and that resist photo-oxidative degradation due to the lack of UV-visible chromophores [15]. However, weathering that occurs in PE during aging can act as chromophores [15].

Oxidation causes hydroxyl and carbonyl groups to form in the chain (see Figure 2) [12,13,15], which can be identified by FTIR. Therefore, this level of degradation can be evaluated using this technique.

The Fourier transform infrared spectrometer was fitted with a golden-gate single-reflection ATR (Attenuated Total Reflection) system, a top plate attached to an optical beam having multiple optical paths. The instrument was operated in double-sided, forward–backward mode. FTIR spectra were collected over the wavenumber range of 5100–600 cm^−1^ at a resolution of 8 cm^−1^ and at a rate of 16 scans per sample.

### 2.5. Determination of Concentration of POPs on Marine MP

A laboratory trial with spiked samples was run to assess the POP adsorption rate according to the kind of plastic involved and its degradation state, simulating the adsorption conditions of chemical pollutants in the ocean.

The concentration of several POPs, which accumulated over different microplastic pellets, was determined. Specifically, 15 organochlorinated pesticides (OCPs), 8 polychlorinated biphenyls (PCBs) and 6 polycyclic aromatic hydrocarbons (PAHs) were analyzed, following an optimized methodology used in previous studies [49]. For POP extraction a solid–liquid–liquid microextraction (µSLLE) was applied, using 10 mL of methanol as extractant with 1 g of MP pellet in an ultrasound bath for 4 min, with the later addition of 100 µL n-hexane based on internal standards. The n-hexane extract was analyzed using GC–MS, following the analytical method described in Appendix A.

POP concentrations were determined at spiked samples for some HDPE pellets at different states of physical and chemical degradation [50]. Seawater spiked with 2 ng·L^−1^ of each of the POPs analyzed was brought into contact with 1 g of each kind of clean MP pellet (any trace of POPs was removed before the experiment) for 24 h, simulating the movement of the sea in an orbital shaker. The variability on POP concentration obtained with commercial virgin, artificial degraded and environmental degraded MP pellets (with different YI values) was studied through the same spiked conditions. Surface/volume of commercial pellets was modified by friction and sun exposure. Moreover, the POP adsorption rate was evaluated according to different levels of the YI (low YI: 25–35%; medium YI: 50–60% and high YI: 70–80%) [30].

## 3. Results and Discussion

### 3.1. ATR-FTIR Analysis

All the sampled pellets were analyzed by ATR-FTIR (between 200–220 pellets), selecting an adequate number of samples for each range of yellowing (at least three pellets per color), initially ordered from a visual point of view (see Figure 3).

Among the marine pellets collected, 120 pellets identified as polyethylene (PE) compounds by ATR-FTIR analysis were selected. Other types of pellets, with a different composition to that indicated, were discarded, thus allowing us to obtain comparable samples.

The samples with more intense yellowing (oranges and browns) obtained lower values of correspondence with the PE reference FTIR spectrum when compared with the data of the spectrum library (values in percentage); nonetheless, they had a correspondence of higher than 70% [10].

In general, spectral indices are not adequate to quantify the aging time of plastics because their behaviors are not linear [10], the chemical properties of MPs change in various ways after aging depending on environmental conditions [22]; however, there is a relationship between the YI and the formation of new carbonyl groups [51].

### 3.2. Yellowness Index (YI) Determination for Microplastic Samples

Following the methodology previously described (Section 2.3), YI values were determined for the 120 PE pellets selected. Results obtained were compared with the color of these samples, by developing a color-grading system based on the 120 samples with values between 20–100% of YI.

Pellets with similar colors obtained similar values of YI (%). This allows for the creation of a reference color palette where the YI value (measured in percentage) is related to the color of the MP pellet (Figure 3, downloadable at https://bit.ly/microplasticYI, accessed on 22 February 2022). This makes it possible to roughly obtain the numerical value of YI of a pellet (for PE composition), without requiring specific analytical equipment, simply comparing the color of the sample with the given color palette.

### 3.3. Polyethylene Photo-Oxidation Process Compared with Its YI Value

The YI value gives a numerical value of the degradation state of a MP pellet, but this also can be evaluated by ATR-FTIR analysis [13].

The characteristic peaks of PE are given by two peaks at wavenumbers 2918 and 2851 cm^−^^1^ (Figure 4), the peaks at 1468 cm^−^^1^ (bending deformation) and 1373 cm^−^^1^ (CH_3_ symmetric deformation), and another peak at 718 cm^−^^1^ (rocking deformation) corresponding to the native bonds present at the polymer [12].

Higher values of YI (%) indicate high degradation of the PE microplastic samples due to the photo-oxidation process suffered by the plastic, generating carbonyl groups that increase the absorbance signal in the range between 1000 and 1400 cm^−^^1^ of the FTIR spectrum, as previously reported [12,15] (Figure 4).

### 3.4. Persistent Organic-Pollutant Adsorption Rate According to Microplastic Degradation Level

MP degradation and fragmentation includes two different processes between others: photo-oxidation of the plastic compound (polyethylene in this case) and the MP surface cracking, which increases the relative surface/volume on MP [52]. Although both processes are related at the marine environment, a lab trial was conducted in this study to evaluate both separately.

Seven different MP pellet samples were evaluated to assess the POPs adsorption rate depending on the degradation state of the MP: varying the relative surface/volume through physical fragmentation processes and varying the photo-oxidation degradation level using MP with different YI values.

The characteristics of each type of MP samples studied were (see Figure 5):(a)HDPE virgin pellets;(b)HDPE virgin pellets subjected to intense friction with metal blades of a food mixer to increase their surface/volume ratio;(c)HDPE virgin pellets subjected to intense friction with clean beach sand to increase their surface/volume ratio;(d)HDPE virgin pellets exposed to 1 year in seawater, in the sun under controlled conditions;(e)Marine MP pellets collected at beaches with low YI value (25–35%);(f)Marine MP pellets collected at beaches with medium YI value (50–60%);(g)Marine MP pellets collected at beaches with higher YI value (70–80%).

**Figure 5 polymers-14-01305-f005:**
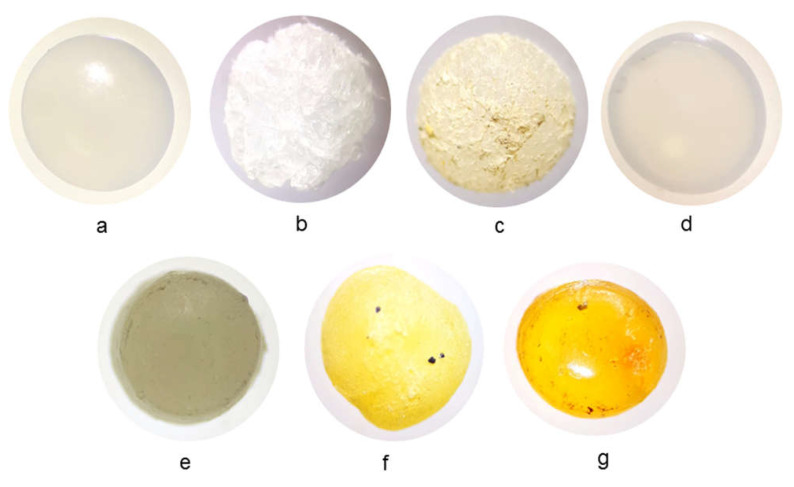
Images of the analyzed pellets under a binocular magnifying glass: (**a**) virgin; (**b**) virgin with high relative surface/volume by artificial friction; (**c**) virgin with high relative surface/volume by sand; (**d**) virgin with sun degradation; (**e**) marine with low YI; (**f**) marine with medium YI; (**g**) marine with higher YI.

Marine MP with different YI values collected at beaches were cleaned before spiking to ensure that they did not have additional POPs apart from those added in this experiment. For the experiment, a clean seawater without POPs (before spiked), confirmed by a seawater blank, was used.

The seven different MP pellet samples under study (1 gram each) were introduced into 100 mL of seawater spiked with the POP mixture (final concentration 2 ng·L^−^^1^). They were left in an orbital shaker for 24 h at 100 rpm to simulate the conditions of ocean movement. After adsorption, desorption was carried out following a solid–liquid–liquid microextraction (µSLLE) with methanol and n-hexane [49] and analyzed by GC–MS to obtain the POP concentration determination [47].

With results equivalent to previous studies [30,40,42,53,54,55], it was observed that the most degraded MP granules (with a high YI value) had a higher rate of adsorption of POPs (higher concentration of POPs under the same adsorption conditions), which is more significant in the case of OCPs in this study (Figure 6). In relation to this, the samples with the highest adsorption rate are those that have undergone a degradation process by photo-oxidation in the marine environment (Figure 6II, right). The high exposure of marine pellets to different environmental factors also causes a physical change in the microplastic surface [12,13,14,22,56] producing surface cracking, thus increasing the surface-to-volume ratio of the MP sample. This increase also influences the POP adsorption rate (in addition to producing the high YI value due to degradation by photo-oxidation) [17,21,36]. Figure 6I (left) shows the comparison of virgin granules and those with a high surface/volume ratio but with a low YI value (as shown in Figure 5b,c).

## 4. Conclusions

There is a proportional relationship between the time that the MP has been exposed to the marine environment and its consequent weathering. The efficient estimation of the weathering and degradation state in marine microplastic samples is a necessity for the laboratories that analyze these types of samples.

MP yellowing is related to its degradation state by solar radiation exposure in marine environments. Although YI (%) is a parameter that allows the quantification of yellowing of a plastic, it is not widely used by the scientific community due to the necessity of specific equipment and because it is a time-consuming procedure.

Identifying a relationship between the color of a sample (visually determined), its yellowing value (YI) and the associated FTIR spectrum allows us to estimate the weathering and degradation state of an MP sample both quantitatively and qualitatively.

The proposed quantified color palette allows an approximate YI value from the MP color for PE pellets to be obtained. Moreover, samples with higher YI values presented an increase in FTIR signals in the area corresponding to the carbonyl groups, which are related to a greater degree of decomposition of PE due to the high exposure of the sample in marine environments. These results make it possible to evaluate, using FTIR analysis, the degradation state of a MP sample.

The aging, degradation, weathering, surface cracking and yellowing of marine MPs over time in the environment are all directly related to the adsorption rate of POPs onto MPs. This has been studied in PE pellets for 15 OCPs, 8 PCBs and 6 PAHs. The increment on POP adsorption rate has two different (but related) reasons: the high relative value of surface/volume due to fragmentation and MP surface cracking; and the increased age of the MPs, which have suffered processes of weathering and yellowing due to the photo-oxidation of the plastic composition. Further studies are needed to explore the effect of plastic oxidation on the adsorption of chemical pollutants in marine environments.

## Figures and Tables

**Figure 1 polymers-14-01305-f001:**
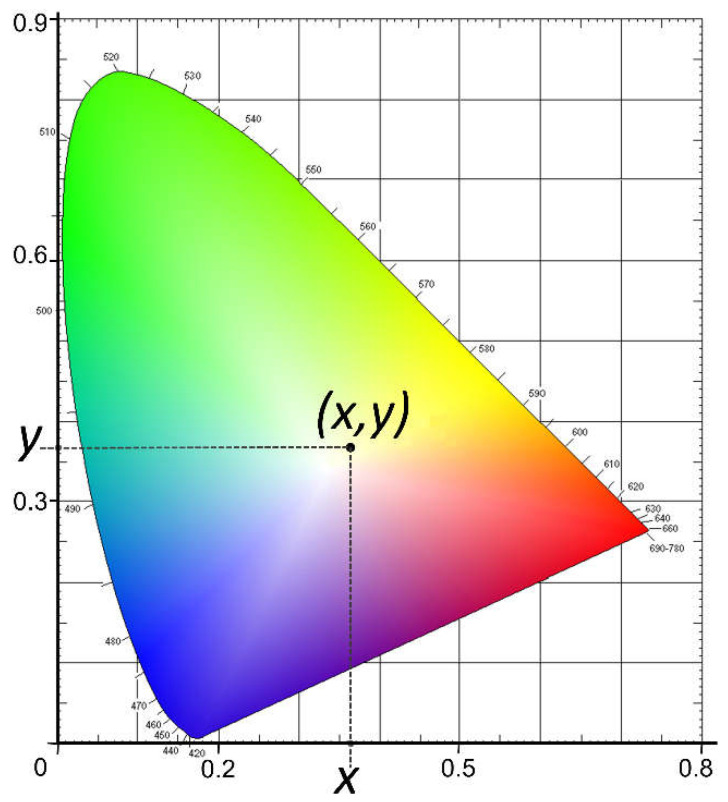
Representation of a CIE three–dimensional color–space diagram.

**Figure 2 polymers-14-01305-f002:**
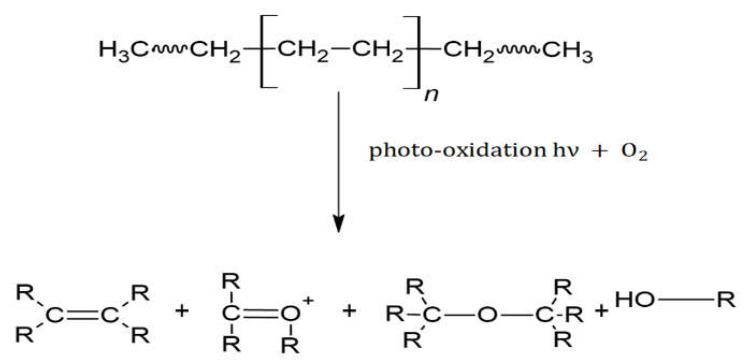
Polyethylene photo-oxidation degradation process (with the carbonyl groups formation).

**Figure 3 polymers-14-01305-f003:**
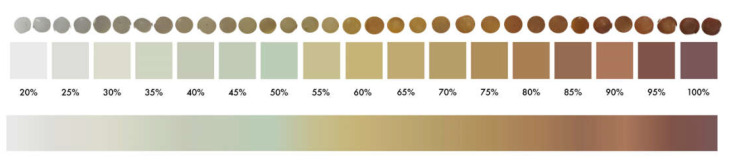
Color palette for Yellowness Index values (%) compared polyethylene pellets classified according to their photo-oxidation degradation process (from a visual point of view).

**Figure 4 polymers-14-01305-f004:**
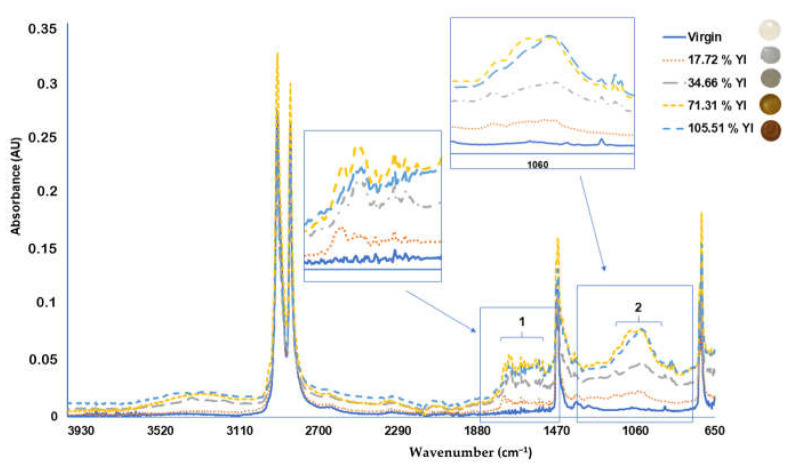
Polyethylene (PE) pellet FTIR-ATR spectrum with different Yellowness Index value; (**1**) PE oxidation with C=O; C=C bonds; (**2**) PE oxidation with C-O-C bonds.

**Figure 6 polymers-14-01305-f006:**
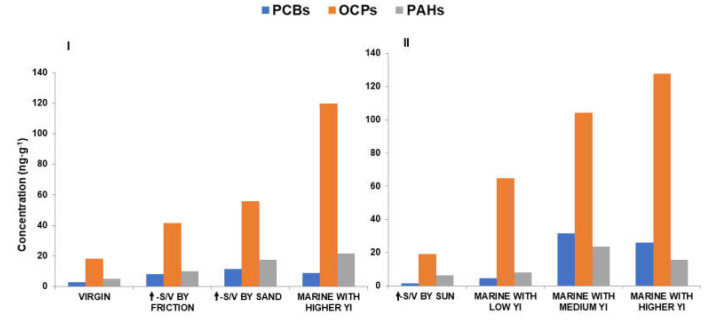
Total concentration of POPs (PCBs, OCPs and PAHs) adsorbed onto PE pellets obtained according to the degradation state of the MP sample. (**I**) Physical degradation and (**II**) chemical degradation.

## Data Availability

Not applicable.

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
