# Peer review of "Yellowing, Weathering and Degradation of Marine Pellets and Their Influence on the Adsorption of Chemical Pollutants"

_polymers, 2022, doi:10.3390/polym14071305_

Round 1

Reviewer 1 Report

The authors address a very relevant and timely topic, with a sound basis in the literature and mostly by a well-designed study. The manuscript is well written, and the work and the results are well presented. Line 124 refers to the objectives of the study. These are not well stated, reducing the readability of the text.

There are minor issues, open for critique: The choice of solvent for the extraction of POPS from the microplastics is ill suited for its task. I doubt the extraction would give a quantitative extraction of all adsorbed POPS. Obviously, the polymer’s vulnerability to the more powerful solvents was behind the choice. I would expect the authors to reveal their thoughts on this.  Also, why was not weight gain used as a measure for the POPS adsorption rate? If it was, then rewrite for a better description of method.

Reviewer 2 Report

The article presents the proposed method for assessing the degree of degradation of microplastics (MP).

The article requires many corrections.

  1.     In the introduction, it would be advantageous to introduce a description of the use of the carbonyl index to describe the degradation of PE or PP.
  2.      Lines 200-204 describe the methodology of FTIR tests. This description should be found in Chapter 2.3.    
  3. Chapter 3.2, lines 206-214, also contain a description of the research methodology,  it should be included in Chapter 2.4. 
  4. Lines 275 - 281 describe the methodology of POPs research, and this description should also be included in the description of the research methodology in Chapter 2.5.
  5. In this article, FTIR infrared spectroscopy was used to describe the MP. However, the authors of the article did not make sufficient use of these results. In the case of PE or PP degradation tests using FTIR, it is very important to calculate the carbonyl index CI). For example, the CI was calculated from the ratio between the integrated band absorbance of the carbonyl (C═O) peak from 1,850 to 1,650 cm − 1 and that of the methylene (CH2) scissoring peak from 1,500 to 1,420 cm − 1. CI is calculated as the ratio of the absorbance or area of the band of the carbonyl (C═O) peak to the absorbance or area of ​​the methylene (CH2) scissoring peak. Other methods of describing CI are also described in the literatu6.     After a precise FTIR analysis, the results of the analysis should be verified.

Reviewer 3 Report

In this manuscript, authors report relationship between weathering status of polyethylene (PE) microplastic pellets and the associated persistent organic pollutants (POPs). It adds a detailed evaluation of weathering of plastic pellets in the marine environment and how the processes are related with the accumulation of POPs in the oceans. As can be expected, more accumulation of POPs (PCBs, OCPs, and PAHs) in PE pellets was observed with increasing Yellowness index (YI). Although the authors concluded that their observation supports a hypothesis that weathering of PE pellets increases the sorptive capacity to those POPs, this is a hasty conclusion because

  • It is well-known that sorption of hydrophobic organic pollutants to (low-density) PE is explained as a partitioning process. Many earlier studies using LDPE as a passive sampler for those POPs confirmed that POPs are almost uniformly distributed in LDPE if sufficient time is allowed.
  • Pellets with greater YI value stayed longer in the environment. This means that they experience longer time to attain phase equilibrium between PE pellets and those POPs in water. Because equilibration time for those highly hydrophobic substances is very long, the accumulation takes long time. POPs are not intentionally added in (LD)PE, thus virgin pellets should contain lower POPs level, while weathered should contain more. The observation cannot be explained by “adsorption rate”. It should be explained by phase equilibrium between PE pellets and water and the uptake kinetics.

Therefore, the manuscript should be extensively revised before it is accepted for publication. Specific comments/suggestions are as follows:

Line 142, 171: Abbreviations, CIE and IS, were used without defining it.

Line 172 and throughout the manuscript: “adsorption rate” is not well defined. Rate means how fast the process occurs, which does not agree with the usage here. Only the accumulated amounts were mentioned.

Line 254 and throughout the manuscript: eight? Or seven? It is confusing.

Figure 6 actually shows that the POPs concentration increases with increasing residence time of PE pellet, suggesting that PE pellets are a sink of POPs from water because distribution equilibrium between PE pellets and water requires decades.
